# Classification of Motor Imagery EEG Signals Based on Data Augmentation and Convolutional Neural Networks

**DOI:** 10.3390/s23041932

**Published:** 2023-02-09

**Authors:** Yu Xie, Stefan Oniga

**Affiliations:** 1Faculty of Informatics, University of Debrecen, 4032 Debrecen, Hungary; 2North University Center of Baia Mare, Technical University of Cluj-Napoca, 400114 Cluj-Napoca, Romania

**Keywords:** motor imagery (MI), electroencephalogram (EEG), data augmentation (DA), convolutional neural network (CNN)

## Abstract

In brain–computer interface (BCI) systems, motor imagery electroencephalography (MI-EEG) signals are commonly used to detect participant intent. Many factors, including low signal-to-noise ratios and few high-quality samples, make MI classification difficult. In order for BCI systems to function, MI-EEG signals must be studied. In pattern recognition and other fields, deep learning approaches have recently been successfully applied. In contrast, few effective deep learning algorithms have been applied to BCI systems, especially MI-based systems. In this paper, we address these problems from two aspects based on the characteristics of EEG signals: first, we proposed a combined time–frequency domain data enhancement method. This method guarantees that the size of the training data is effectively increased while maintaining the intrinsic composition of the data. Second, our design consists of a parallel CNN that takes both raw EEG images and images transformed through continuous wavelet transform (CWT) as inputs. We conducted classification experiments on a public data set to verify the effectiveness of the algorithm. According to experimental results based on the BCI Competition IV Dataset2a, the average classification accuracy is 97.61%. A comparison of the proposed algorithm with other algorithms shows that it performs better in classification. The algorithm can be used to improve the classification performance of MI-based BCIs and BCI systems created for people with disabilities.

## 1. Introduction

A brain–computer interface (BCI) is a system that facilitates communication between the human brain and external devices (such as a computer or other electronic devices) without the need for any intermediaries. It allows the user to control the computer or smart device directly through signals generated by the brain without the involvement of peripheral organs and muscles. BCI research has significant theoretical, military, medical, and recreational value. BCI research has considerable value across a wide range of fields [1,2,3,4]. Theoretically, it has the potential to offer unprecedented insight into the functioning of the human mind. For military purposes, it can enhance soldiers’ efficiency in the field. In the medical realm, it could potentially be used to treat certain conditions, such as paralysis, that have previously been considered untreatable. Finally, it enables the creation of captivating leisure experiences, allowing users to delve into virtual settings and interact with them in unprecedented ways. Electroencephalography (EEG) is a widely utilized signal for building BCI. It offers many advantages, such as being cost-effective, non-invasive, and portable [5]. At the same time, we face the challenge of the high dimensions EEG signal being too weak and the signal-to-noise ratio being low [6]. Furthermore, MI-EEG is a non-linear and unstable signal, which means that its parameters (e.g., mean and variance) change over time [7].

Several scholars have recently used convolutional neural networks to feature the extraction of EEG signals. This technique significantly decreases the number of connections and parameters in a deep network while using the pertinent geographical or temporal links between nearby data points to infer helpful features for a particular machine learning job. Ref. [8] proposed a frequency complementary map selection (FCMS) scheme based on augmented CSP (ACSP) for convolutional neural networks (CNNs) to reduce the dependence of feature mapping across frequency bands. However, this method fails to take full advantage of the temporal information of EEG. Ref. [9] used short-time Fourier transformation (STFT) to map the EEG signals to image signals and then used CNN and automatic stacking encoding (SAE) to extract features and classify them. In [10], a CNN is used to learn the temporal information of EEG based on envelope representations filtered by Hilbert transforms. Since the envelope signal is a low-frequency signal, it can be down-sampled, thus reducing the dimensionality of the data. CNNs and other deep learning algorithms need a large amount of data in order to achieve competitive performance on the target task. If we lack sufficient data to train, training deep learning models can overfit certain features to a specific training set, thus limiting the generalizability of the model. The model’s limitations lie in its inability to correctly predict or comprehend new data, due to only being trained on a particular set of data. In order to increase the accuracy of the model, we need more data and ensure that it is representative of the data we expect to encounter in the future. This will help the model to generalize better and provide more accurate predictions.

Due to the experiments’ intricate procedures and rigid guidelines to acquire MI-EEG signals, it is challenging to obtain enough training data to classify MI-EEG signals [11]. For instance, blinking and body trembling can both produce sudden spikes in the EEG signal; these are known as eye movement artifacts and muscle movement artifacts, respectively [12]. Additionally, acquiring a valid sample requires several preparation processes, following the pointing arrows on the screen and imagining the movement followed by a short pause. All this requires the subject’s attention during this period, but as the experiment time increases, it becomes challenging to maintain attention all the time. As a result, it is incredibly difficult to collect enough good-quality MI-EEG signals for the training process. Researchers usually employ data augmentation (DA) technology to address general overfitting issues. Through data transformation or oversampling, DA techniques enhance the volume of existing data sets [13]. It has been shown that DA is successful in a variety of fields, including target recognition and image processing [14]. Typically, enhanced data are produced using two techniques. The first method involves employing geometric transformations, while the second entails adding noise to the existing training data. Ref. [15] proposed a multi-input CNN to classify multi-channel MI-EEG Signals. They rotated and flipped the MI-EEG image to achieve the purpose of increasing the data volume. Ref. [11] proposes a DA method that combines rotating images and adding noise. The EEG, in contrast to images, is a collection of boisterous, particularly relevant (in time and space), non-stationary time series from various electrodes. Geometric transformations do not directly apply to EEG data due to the risk of distorting time-domain features [16]. Ref. [17] applied the empirical mode decomposition (EMD) to create new artificial EEG frames, followed by transforming all EEG data into tensors as inputs of the neural network by complex Morlet wavelets. EMD separates the EEG signal into its intrinsic mode functions (IMFs), which are usually narrowband, and the resulting IMFs can be analyzed individually to extract information about specific frequency bands, which are thought to be related to different cognitive processes. DA methods [13,18,19] appropriate for EEG signals are crucial to be investigated at this point. Overall, designing a CNN suitable for feature extraction and classification for a segment of highly complex and unstable EEG signals is a great challenge, while using CNNs for signal classification has the potential demand for a large amount of data and system robustness. In this paper, we address these problems from two aspects based on the characteristics of EEG signals: first, we proposed a combined time–frequency domain data enhancement method. Our proposed DA method differs from current approaches in that it operates in both the time and frequency domains to transform the data, which ensures the effectiveness of increasing the size of the training data, while still preserving the inherent structure of the data. Second, we designed a parallel inputs CNN followed by raw image and continuous wavelet transform (CWT) transforming EEG image as inputs and conduct classification experiments on a public data set to verify the effectiveness of the algorithm. This model not only emphasizes the main features of the original data, but also preserves other valuable features of the original data.

## 2. Methods

CWT and CNNs are both used in EEG analysis to extract essential features and patterns from EEG signals. The CWT provides a time–frequency representation of EEG signals, capturing both frequency content and temporal evolution. This is important in EEG analysis because EEG signals are non-stationary and transient in nature, and the CWT can effectively capture these features. CNN, on the other hand, is a type of deep learning model that is well-suited for image and signal analysis, making them a useful tool in EEG analysis. By learning complex patterns and relationships in EEG signals, CNNs can identify important EEG features, such as spikes and oscillations, that are indicative of brain activity and neurological conditions. By leveraging the advantages of both CWT and CNNs, EEG analysis can take advantage of the time–frequency representations provided by CWT, as well as CNNs’ ability to identify intricate patterns in EEG signals. This leads to improved EEG analysis and a better understanding of brain activity and neurological conditions.

### 2.1. Continuous Wavelet Transformation (CWT)

CWT is a popularly used method for the time–frequency analysis of signals. It was proposed by Morlet and Grossman in 1987. CWT technique provides a time–frequency representation of signals by decomposing a signal into wavelets of different scales, which are functions that are used to analyze the different frequency components of the signal. This allows for a more complete analysis of signals compared to traditional frequency analysis methods, which only provide a single frequency representation of the signal at a specific point in time. The CWT has proven to be an effective tool for analyzing EEG signals with respect to their frequency and temporal characteristics, which is crucial for understanding brain activity and diagnosing various neurological conditions [20,21,22].

The expression of the continuous wavelet transform is shown in Equation (1) [23].
(1)$$w_{s}\left(a,\tau \right)=a^{\frac{1}{2}}\int s\left(t\right)\phi^{*}\left(\frac{t-\tau }{a}\right)dt,$$
where *s*(*t*) is the input signal, *a* is the scaling of the wavelet transform, *ϕ* is the wavelet basis function, and *τ* is the time offset. There are five commonly used wavelet basis functions: Morlet wavelet, Mexican Hat wavelet, Haar wavelet, Daubechies wavelet and SymN wavelet clusters. We choose the Morlet wavelet as the wavelet basis function. Its time-domain expression is as follows:(2)$$\phi \left(t\right)={\left(\frac{2}{\pi T^{2}}\right)}^{\frac{1}{4}}exp\left(-\frac{t^{2}}{T^{2}}+jw_{c}t\right),$$

The expression of frequency is:(3)$$\phi \left(w\right)={\left(\frac{T^{2}}{2\pi }\right)}^{\frac{1}{4}}exp\left(-\frac{{\left(w-w_{c}\right)}^{2}}{4T^{2}}\right),$$

By analyzing the data, *T* and $w_{c}\;$of the wavelet function are determined. We use CWT to preprocess the original signal and then use the mapped time–frequency domain image as one of the inputs to the proposed CNN.

### 2.2. CNN

Deep learning is an essential aspect of machine learning—an emerging field that keeps moving forward. In many fields, deep learning has gradually become a pioneer, attracting the attention of numerous scholars [24,25,26,27]. In deep learning, CNNs are a widely used neural network model with multiple applications across various fields [28,29,30]. Meanwhile, there is also relevant research progress in BCI systems [31,32]. Multiple convolutional and pooling layers can be combined in the network structure’s middle, followed by fully connected layers.

The heart of CNN is the convolutional layer and its primary function of the convolutional layer is to extract features from the input signal. The ability of the convolutional layer to perform the relevant convolutional operations on the input signal is crucial. Convolutional kernels, also known as filters, can exist in more than one in the framework of a single convolutional layer. The weight parameters and bias of the convolution kernels can be changed during the training of the neural network. Matrix multiplication principles can be used by convolutional operations to generate feature mappings from the input to the output. The position of the neural element in the feature map output by the kth convolution kernel is assumed as (*m*, *n*). The output is shown in Equation (4)
(4)$$y_{m,n}=f\left(w{\left(i\right)}_{k}*I\left(m,n\right)+b\right),$$
where $I\left(m,n\right)$ is the input data, b is the bias, $w{\left(i\right)}_{k}$ is the *k*th convolution kernel of the ith layer, f is the activation function of the nerve element, and its common form contains tanch, sigmoid, and rectified linear unit (ReLU) [33].
(5)$$tanch:f\left(x\right)=\frac{1-e^{-2x}}{1+e^{-2x}},$$
(6)$$sigmoid:f\left(x\right)=\frac{1}{1+e^{-x}},$$
(7)$$\;\;\;ReLU:f\left(x\right)=\max\left(0,x\right),$$

On the side, subsequent connections to tighter layers enable the extraction of more distinct features. The features of high-dimensional input data can be continuously extracted by combining and superimposing multiple convolutional layers so that more advanced abstract features can be obtained from the signal to the greatest extent. In addition to the convolution operation, the convolution layer also includes the processing of operations such as padding and stride, and the associated computational process is more complex.

Pooling, also known as the sampling layer, is the process of reducing the input image’s length and width sizes. Following the convolutional layer, the pooling layer can perform the operation of downsampling to extract local features. The pooling layer allows reducing the number of parameters of the network, which means that the computational effort and complexity of the model are also decreased. At the same time, this can be very robust to small errors in the data and improve the overfitting problem of the network model. The pooling layer has two primary pooling operations. By maximum pooling, the target region’s feature value is determined by the largest element value in the region. The maximum pooling operation emphasizes more local features and optimizes the small errors generated by the convolution layer. By averaging pooling, similarly, the target region’s feature value is determined by the average value of elements in it. In addition to retaining more adjacency information, the averaging pooling operation improves the region error caused by the size of the convolution kernel. The maximum and average pooling expression is shown in Equations (8) and (9). We assume that the size of the pooling kernel is (*N* ∗ *N*).
(8)$$Max\;pooling:f\left(x\right)=\max\left(x_{\left[m,m+N\right],\left[n,n+N\right]}\right),$$
(9)$$Average\;pooling:f\left(x\right)=\frac{1}{N*N}\sum_{m=m_{1},\;n=n_{1}}^{m=m_{1+N},\;n=n_{1+N}}x_{m,n},$$

The fully connected layer is generally treated as the concluding component of the CNN structure. The feature maps produced by intermediate layers are converted into vector format by the feature space transformation when it passes through the fully connected layer, which combines the previously extracted features for use in matrix multiplication. The fully connected layer is responsible for transforming the spatial high-dimensional features that have already been extracted by the CNN and concluding the overall learning process with non-linear mapping.

The better robustness of the CNN network model and its strong generalization performance are mainly attributed to the use of design ideas such as sparse connectivity of convolutional layers, weight sharing, sampling of pooling layers and non-linear mapping of fully connected layers. In traditional recognition studies, a large number of samples are analyzed and processed as a requirement for effective feature extraction and classification recognition. In short, CNN models are automatically trained on the intrinsic features of the signal through convolutional operations and other related operations [34]. CNN has not only excellent feature extraction effects but also has strong interpretability in its network model structure.

### 2.3. Proposed CNN Structure

Event-related desynchronization (ERD)/event-related synchronization (ERS) phenomena typically occur between 8 and 32 Hz in MI-EEG signal tasks, according to related studies [35,36]. In [37], the researchers designed a CNN model with a parallel structure using three different frequency bands as input. They also designed a mixed convolutional scale in their model by assigning each frequency band to the convolutional layer corresponding to three different scales of the convolutional kernel. This adoption of a parallel approach to extracting features at multiple scales improves the accuracy of MI EEG signal classification. In [38], the authors proposed a parallel multiscale filter bank convolutional neural network for MI classification. They used time–domain images as input and then used four kinds of different time–scale convolutional kernels for feature extraction to improve the performance, robustness, and migration learning of the model. The most popular application of CNN is to use either time domain signals or frequency domain signals [9] as input. Since EEG models have both continuous and complex variations, it is hardly possible for the model to extract enough features from only frequency or time domain dimension, a multi-input time–frequency CNN structure is proposed in this paper.

As the EEG signal has temporal, frequency, and spatial information, the proposed CNN is divided into two parts in order to extract more comprehensive temporal and spatial features. As shown in Figure 1, the input to the left half of the model is a time-domain image. This section focuses on modifying the size of the convolution kernel of the original model [31]. The first and second convolutional layers of the network are 1-D convolutional layers with kernel size 1 × 3 and 10 × 1, and stride step size 2 × 1 and 2 × 1, which are used to learn the spatial information and temporal features between each channel, respectively. The right half of the model input is the image after CWT mapping. The overall and local features of the EEG signal may be more clearly evident following the wavelet transform of the original signal, allowing the network to extract features more effectively [34]. The two convolutional layers consist of 32 and 64, convolutional kernels size 5 × 5, and stride step size 2 × 2 and 2 × 2, respectively. The 2-D feature maps extracted from the left and right parts are expanded into a 1-D vector by a fully connected layer. Then, we concatenate all the 1D features from the two branches into a 1D vector and use the vector as the input of the classifier. In this work, we use the SoftMax function as the classifier. ReLUs are used as the activation function in the proposed model because it increases classification accuracy while speeding up convolutional neural network learning. To prevent overfitting of the neural network, L2 regularization is used. In the training, the L2 regularization parameter is set to 0.01, and the Adam optimizer is employed as the optimization method. The initial learning rate is set to 0.1, and the learning rate is automatically optimized. Batch size and number of epochs are set to 64 and 100, respectively. The early stop mechanism is also applied in the training phase. The training process is stopped when the validation loss stops decreasing within 5 epochs. This is performed to prevent the model from continuing to fit the training data and instead keep the best-performing weights that generalize well to new data.

CNN’s pooling layer can simplify the network parameters, but some useful features might be lost. In [39], a method to classify MI-EEG signals using a simplified CNN was proposed. In this paper, the pooling layer is removed from the standard CNN to optimize the network structure and prevent the loss of effective features.

### 2.4. Data Augmentation

The EEG signals collected by each electrode are commonly considered as 1-D data, in contrast to the computer vision field, where images are often rotated, cropped, deformed, scaled, and subjected to other types of DA methods that frequently use 2-D data. The two main categories of the current EEG signal data enhancement techniques are data transformation [18,40,41] and noise addition [16,42].

According to the analysis of the essential features of EEG signals, it can be seen that EEG signals have relatively obvious characteristics in both time and frequency domains. The DA algorithm is designed in the time and frequency domains while maintaining the original characteristics of the EEG signal as much as possible in this work.

The proposed method will be divided into two steps: time domain transformation and frequency domain transformation, as shown in Figure 2 and Figure 3.

Sample 1, sample 2, and sample 3 are three samples of the same class that were randomly selected. We randomly capture a period of 1 s of data from sample 2 to replace the data at the same time position in sample 1. Figure 1 shows a time domain sample generation. Do the same for the test set;The artificial time-domain EEG sample and sample 3 are divided into two frequency bands, 7–13 Hz and 14–30 Hz, after band-pass filtering, and then a frequency band of sample 3 is exchanged with the corresponding frequency band of the artificial time-domain EEG sample to reconstruct the time–frequency EEG signal.

## 3. Results

All the experimental results are obtained using a computer equipped with an Intel Core i5-7300HQ CPU and a GeForce GTX 1050Ti GPU.

### 3.1. Database

Since 2001, several international BCI competitions have been held to deliver scientists working in this area with reliable data sources and uniform standards of detection algorithms. To test the performance of the proposed methodology, we selected a public data set (2008 BCI Competition IV Data Set 2a [43]) for this purpose.

A diagram of the acquisition process for this data set can be seen in Figure 4. The complete acquisition process of a single EEG signal is divided into three parts. The first 3 s of a trial is the preparation phase. The directional arrow prompts appear on the screen and are accompanied by an audible alert. Subjects need to follow the on-screen prompts to start the motor imagery from the 3rd to the 6th second. After that, the subjects rested and waited for the cue to start the next trial. We intercepted from 3.5 s to the end of the 6th second, considering that the subject might not be able to respond when the cue first appeared, which means that 625 data points were obtained as an original sample in our experiment. The data format for each raw sample is 625 × 22, where 625 is 2.5 s of data points (250 Hz) and 22 means there are 22 electrodes.

Data are collected from each subject in a total of 576 trials. We repeated the proposed DA process 1000 times to obtain 1000 new samples. The final number of samples for each subject increased from the original 576 to 1576.

### 3.2. Analysis of DA Methods

Figure 5 and Figure 6 show the spectral power comparison plots before and after the signal augmentation, respectively. The red line is the spectrum of the original data, and the blue line is the spectrum after the augmentation. Further, the spectrum of C3, Cz, and C4, which are more closely imagined with the motion, are shown as examples. By comparison, it was found that the characteristics of the original EEG signal were largely retained before and after our augmentation method. The proposed DA method focuses on increasing the energy of the replaced band while keeping the spectral ratio of the signal to its replaced band relatively constant. This process can be used to improve the signal-to-noise ratio and to gain a better explanation of the signal’s characteristics.

### 3.3. Performance Evaluation Metrics

In this section, we introduce several practical metrics to evaluate the performance of the proposed method. The accuracy Acc is calculated as:(10)$$Acc=\frac{T_{p}+T_{N}}{T_{p}+T_{N}+F_{p}+F_{N}},\;Acc\epsilon R\left[0,1\right],$$
where $T_{p}$, $T_{N}$, $F_{p},$ and $F_{N}\;$represent true positives, true negatives, false positives, and false negatives, respectively.

The other three important metrics, *Precision*, *Recall*, and *F*1 score, can be expressed as:(11)$$Precision=\frac{T_{p}}{T_{p}+F_{p}},$$
(12)$$Recall=\frac{T_{p}}{T_{p}+F_{N}},\;a\epsilon R\left[0,1\right]$$
(13)$$F1=2*\frac{Recall+Precision}{Recall*Precision},\;a\epsilon R\left[0,1\right]$$

### 3.4. Determining the Length of Data Segments

Data segmentation and overlap are common ways to increase the amount of data, which can significantly improve classification accuracy [44]. It can be extremely beneficial in terms of reducing resource consumption during network calculational operations. Not only does it decrease the size of the input, but it also reduces the amount of data that needs to be transmitted and stored during the training and inference process, resulting in improved performance and reduced costs. Consequently, these methods can be valuable optimization tools for us. We used a 50% overlap window for the sample, which means that 50% of the data are the same as in the previous segment. The next thing we determined was the scale of the segmentation. Figure 7 shows the accuracy comparison of different size segments, where the 0.5 s window has the worst classification accuracy, and the 2 s and 3 s windows have more similar accuracy. This means that the data of the 2 s fragment already contain a sufficient amount of MI information. To reduce the resources used, subsequent experiments all use a 2 s time window.

### 3.5. Performance of the Proposed Model

To demonstrate the superiority of the proposed method, we compared our results with some state-of-the-art models. Table 1 introduces the presentation of comparative literature.

We retrained two high-cited models (VGGNet and EEGNet) to evaluate their performance using the same dataset. The comparison results are shown in Table 2. In addition, research methods with the suffix (_A) indicate the use of DA techniques.

According to the proposed method, the accuracy is improved to 97.61%, the highest among the compared methods. In addition, Our DA method can achieve varying degrees of accuracy improvement, with average accuracy improvements of up to 4.41% and 11.15% for the individual subject.

The compared results with other articles [37,46,47,48] are shown in Table 3. It can be seen that our method has a relatively high classification accuracy among the state-of-the-art methods. The proposed method shows a large improvement in the accuracy of subject 2 (up to 18.01% increase) and subject 6 (up to 19.45% increase). Most specifically, compared with Table 2 and Table 3, there is no significant difference in classification performance between subjects (the maximum difference between each subject is 5%, which reaches 32% in VGGNet and 12% in DWT-CNN). It indicates that the method in this study overcomes the effect of individual EEG signal variability and can automatically learn the unique EEG signal activity patterns of different subjects in four imaginary states.

Alternatively, there are various other metrics that can be employed to evaluate the performance of our proposed model. The confusion matrices for the proposed model are given in Figure 8 and Figure 9. Table 4 provides a comprehensive analysis of the precision and recall values of our proposed model on the BCI Competition IV dataset 2a.

## 4. Conclusions

As a result of the complexity of EEG signals, the automatic classification accuracy is usually not high and individual differences are significant in the case of limited training data. Another challenge is that existing CNN-based methods for EEG MI classification use a single domain to extract EEG features. This leads to limited classification since it requires simultaneous decoding of the time domain, frequency domain, and spatial information of EEG signals. We propose a hybrid CNN architecture with a data enhancement approach. The accuracy of EEG motor imagery classification is improved by using more different dimensional images as input to the CNN while increasing the amount of data.

The optimal window size may vary from dataset to dataset, so determining the size of the sliding window can improve the accuracy appropriately. For the 2008 BCI competition IV 2a dataset, the window size of 2 s meets our requirements, which not only reduces the consumption of computational resources but also has an acceptable accuracy rate. Our DA method has been proven to be highly effective in improving the accuracy of training VGGNet, EEGNet, and our proposed model. Overall, our proposed MI-EEG image classification method achieves an average accuracy of 97.61%. The improvement of this method is the design of two different scales of CNN for both time domain and CWT mapping maps, which makes the feature extraction more comprehensive. It also has the highest average classification accuracy compared to several other methods. This indicates the potential of our DA technique in helping to improve the performance of machine learning models. We believe this technique has the potential to be applied to other machine learning and MI-EEG analyzing tasks, and we look forward to further exploring its capabilities. In the future, we aim to design a specialized inference accelerator for this model that can be easily integrated into reconfigurable devices such as field programmable gate arrays (FPGAs). However, this goal presents several design challenges, particularly with the parallel CNN architecture. Both the CNN and CWT mapping operations require a significant amount of computational resources, which may result in insufficient memory or slow inference speed for the devices used. To achieve our goal, we need to overcome these difficulties by finding innovative solutions that allow us to effectively utilize the available resources while ensuring optimal performance.

## Figures and Tables

**Figure 1 sensors-23-01932-f001:**
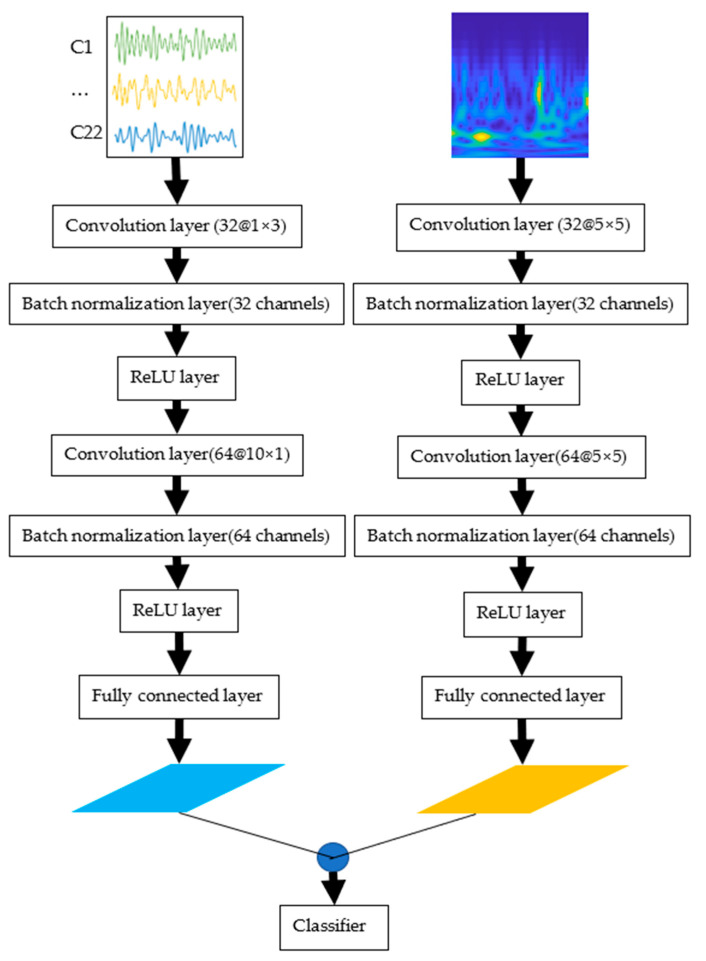
The proposed CNN framework.

**Figure 2 sensors-23-01932-f002:**
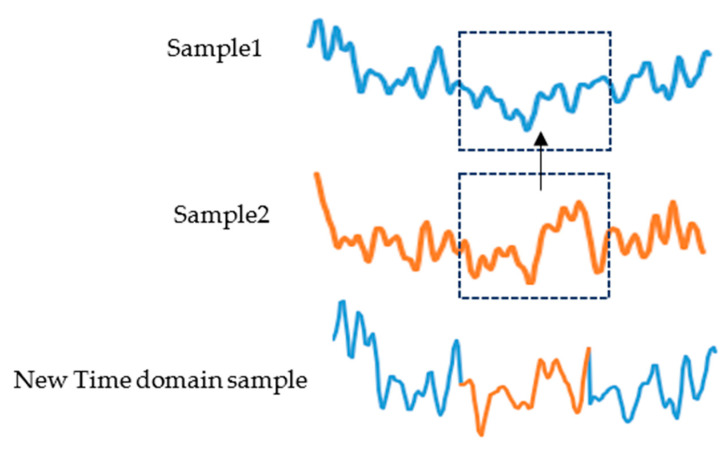
An example of time domain sample generation.

**Figure 3 sensors-23-01932-f003:**
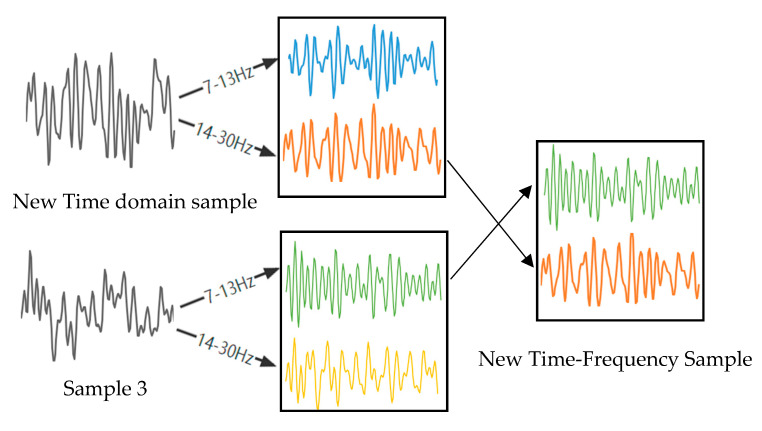
An example of time–frequency domain sample generation.

**Figure 4 sensors-23-01932-f004:**
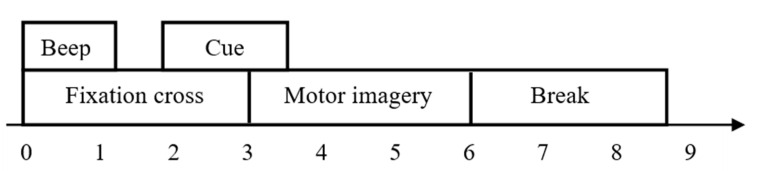
Timing scheme of the paradigm.

**Figure 5 sensors-23-01932-f005:**
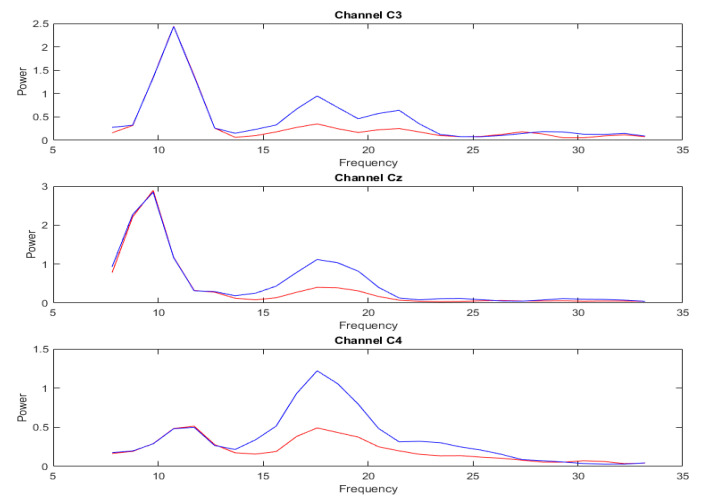
The spectrum of subject 1 imagined the left-hand movement before and after enhancement, in which the 14–32 Hz band of the original signal was replaced. The signal depicted by the red line represents the pristine data, while the signal represented by the blue line corresponds to the output obtained following DA.

**Figure 6 sensors-23-01932-f006:**
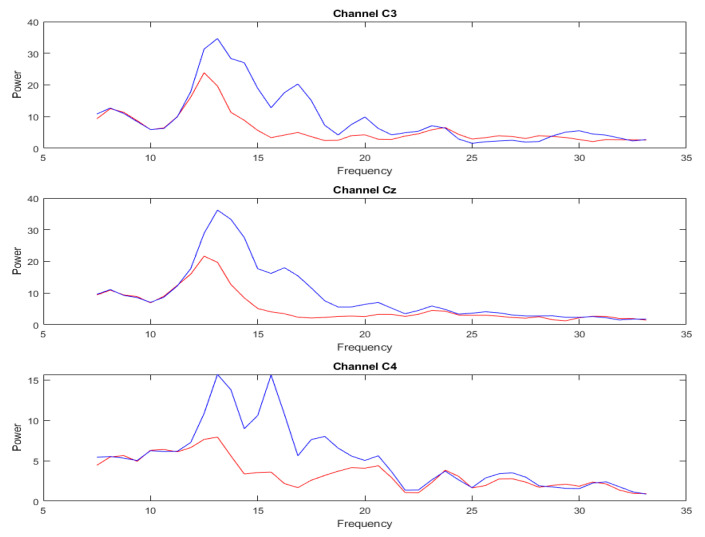
The spectrum of subject 1 imagined the left-hand movement before and after enhancement, in which the 7–14 Hz band of the original signal was replaced. The signal depicted by the red line represents the pristine data, while the signal represented by the blue line corresponds to the output obtained following DA.

**Figure 7 sensors-23-01932-f007:**
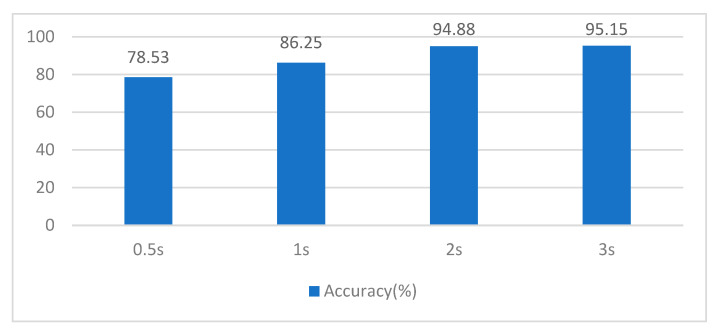
Comparison of different length segments.

**Figure 8 sensors-23-01932-f008:**
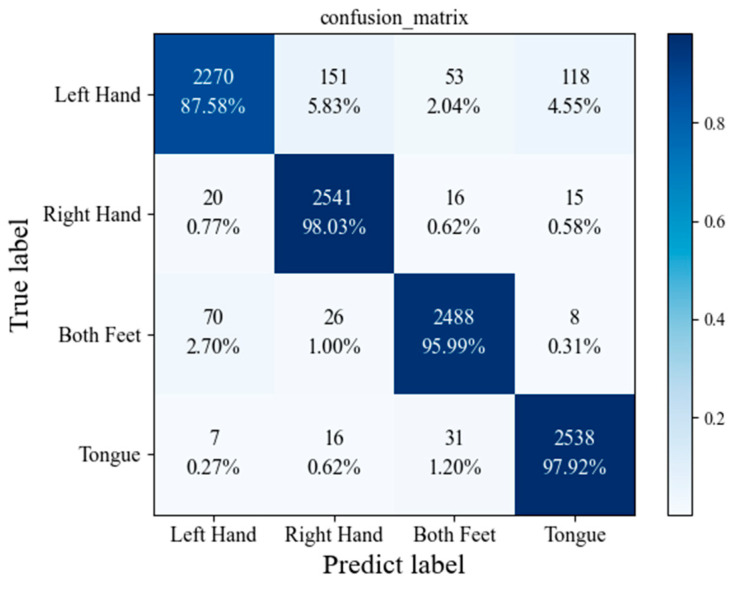
Confusion matrix for the proposed model.

**Figure 9 sensors-23-01932-f009:**
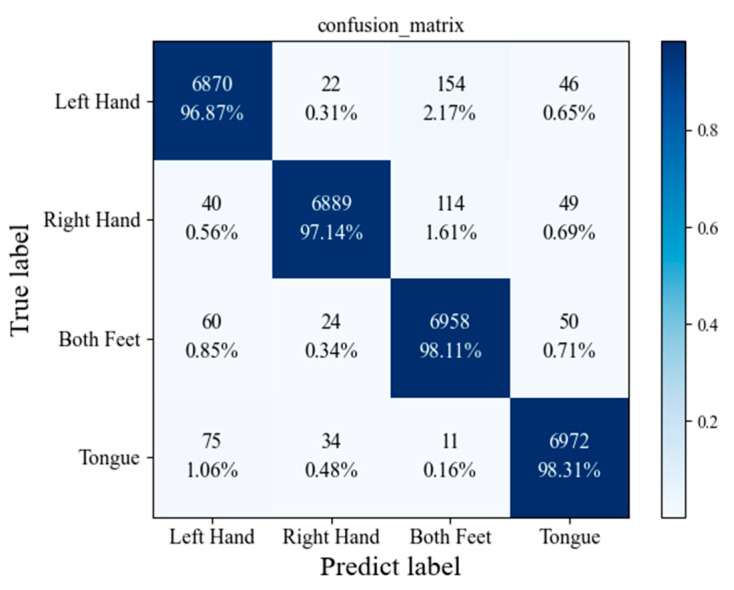
Confusion matrix for the proposed model with DA.

**Table 1 sensors-23-01932-t001:** Contribution to the comparative literature.

Literature	Classifier	Contribution
VGGNet [45]	CNN	Using VGGNet for MI dataset training. Better efficiency and accuracy compared to SVM, ANN, and traditional CNN.
EEGNet [26]	CNN	Introduce EEGNet, a compact convolutional neural network for EEG-based BCIs. Demonstrate three different approaches to visualize the contents of a trained EEGNet model to enable interpretation of the learned features.
HS-CNN [37]	CNN	Different convolution kernel scales are designed for EEG signals in different frequency bands, and convolution operations are performed in the temporal and spatial dimensions using one-dimensional convolution kernels. Identically labeled samples are segmented and then swapped and recombined, then divided by frequency bands, and different segments of the same frequency band are swapped and recombined.
PSO-CNN [46]	CNN	The signal is input to the CSP filter, while the PSD of different frequency bands is calculated, and the particle swarm optimization algorithm is introduced for feature optimization. CNN has only one convolutional layer.
CNN-LSTM [47]	CNN+LSTM	Using the Filter Bank Common Spatial Pattern technique Hybrid network to extract multimodal features.
DWT-CNN [48]	CNN	Sliding window for data enhancement, calculating PSD for optimal frequency bands. Decompose 3 × 3 convolution kernels into 4 × 1 or 2 × 1.

**Table 2 sensors-23-01932-t002:** Compared results with EEGNet and VGGNet.

Subject No.	Acc (%)					
	VGGNet	VGGNet_A	EEGNet	EEGNet_A	Our	Our_A
1	79.32	85.36	88.21	91.33	97.23	98.61
2	69.01	77.22	70.35	72.58	94.77	98.29
3	91.11	92.67	92.40	93.14	94.12	97.99
4	68.32	66.12	81.26	85.65	95.22	97.1
5	59.00	70.15	85.77	83.85	93.25	96.72
6	70.32	71.39	73.12	76.98	92.36	96.37
7	82.55	85.73	93.10	98.54	94.8	97.69
8	79.33	88.52	90.73	89.32	95.75	97.22
9	70.74	72.29	91.05	89.99	96.38	98.51
Mean	74.41	78.82	85.11	86.82	94.88	97.61

**Table 3 sensors-23-01932-t003:** Compare the table of the proposed method with state-of-the-art methods.

Subject No.	Acc (%)					
	HS-CNN	PSO-CNN	CNN-LSTM	DWT-CNN	Our	Our_A
1	90.07	93.30	98.82	98	97.23	98.61
2	80.28	84.59	98.64	98	94.77	98.29
3	97.08	91.68	96.92	95	94.12	97.99
4	89.66	84.55	96.50	96	95.22	97.1
5	97.04	86.54	92.75	86	93.25	96.72
6	87.04	76.92	91.84	92	92.36	96.37
7	92.14	94.03	95.07	94	94.8	97.69
8	98.51	93.20	95.25	93	95.75	97.22
9	92.31	92.24	99.23	98	96.38	98.51
Mean	91.57	85.56	96.13	94	94.88	97.61

**Table 4 sensors-23-01932-t004:** Precision and recall scores for the proposed model.

Metrics	Our	Our_A
Precision (%)	94.93	97.62
Recall (%)	94.88	97.61
F1-Score	0.95	0.98

## Data Availability

Data available on request from the authors.

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
