# Peer review of "Classification of Motor Imagery EEG Signals Based on Data Augmentation and Convolutional Neural Networks"

_sensors, 2023, doi:10.3390/s23041932_

Round 1

Reviewer 2 Report

This manuscript proposed a novel data augmentation method for the classification of motor imagery EEG signals. The experiments are interesting and could be potentially accepted for publication. Some essential issues should be addressed before further process.

(1)   English grammar and spelling must be improved throughout the manuscript.

(2)   In Line109, for recent achievements using deep learning technology, some high-cited papers (https://doi.org/10.1109/ACCESS.2019.2940527) are recommended to be added in the Introduction.

(3) The discussion and limitations of this manuscript are suggested to be further analyzed and presented.

Reviewer 3 Report

The paper proposes a convolutional neural network methodology with a combined time-frequency domain data enhancement approach for the classification of motor imagery electroencephalography signals. The following issues need to be addressed for the publication of the paper in the journal Sensors:

1) The repetitive wording in line 61 as “…we need more data and ensure to ensure that it…”, and in line 68 as “Additionally, In addition …” should be corrected.

2) References should be provided for the statements in lines 64-68.

3) The following paper, which has a very similar title with the manuscript, was not cited in the references:

J. F. Hwaidi and T. M. Chen, "Classification of Motor Imagery EEG Signals Based on Deep Autoencoder and Convolutional Neural Network Approach," in IEEE Access, vol. 10, pp. 48071-48081, 2022, doi: 10.1109/ACCESS.2022.3171906.

The following papers are also closely related to the manuscript and should be cited:

·         T. H. Shovon, Z. A. Nazi, S. Dash and M. F. Hossain, "Classification of Motor Imagery EEG Signals with multi-input Convolutional Neural Network by augmenting STFT," 2019 5th International Conference on Advances in Electrical Engineering (ICAEE), Dhaka, Bangladesh, 2019, pp. 398-403, doi: 10.1109/ICAEE48663.2019.8975578.

·         Zhang, K.; Xu, G.; Han, Z.; Ma, K.; Zheng, X.; Chen, L.; Duan, N.; Zhang, S. Data Augmentation for Motor Imagery Signal Classification Based on a Hybrid Neural Network. Sensors 2020, 20, 4485. https://doi.org/10.3390/s20164485

·         Z. Zhang et al., "A Novel Deep Learning Approach With Data Augmentation to Classify Motor Imagery Signals," in IEEE Access, vol. 7, pp. 15945-15954, 2019, doi: 10.1109/ACCESS.2019.2895133.

·         M. Parvan, A. R. Ghiasi, T. Y. Rezaii and A. Farzamnia, "Transfer Learning based Motor Imagery Classification using Convolutional Neural Networks," 2019 27th Iranian Conference on Electrical Engineering (ICEE), Yazd, Iran, 2019, pp. 1825-1828, doi: 10.1109/IranianCEE.2019.8786636.

·         Z Dokur, T Olmez, Classification of motor imagery electroencephalogram signals by using a divergence based convolutional neural network, Applied Soft Computing, 2021, 113, 107881, https://doi.org/10.1016/j.asoc.2021.107881.

·         Z Khademi, F Ebrahimi, H M Kordy, A transfer learning-based CNN and LSTM hybrid deep learning model to classify motor imagery EEG signals, Computers in Biology and Medicine, 2022, 143, 105288, https://doi.org/10.1016/j.compbiomed.2022.105288.

·         W Ma, H Xue, X Sun, S Mao, L Wang, Y Liu, Y Wang, X Lin, A novel multi-branch hybrid neural network for motor imagery EEG signal classification, Biomedical Signal Processing and Control, 2022, 77, 103718, https://doi.org/10.1016/j.bspc.2022.103718.

·         Shajil, N., Mohan, S., Srinivasan, P. et al. Multiclass Classification of Spatially Filtered Motor Imagery EEG Signals Using Convolutional Neural Network for BCI Based Applications. J. Med. Biol. Eng. 40, 663–672 (2020). https://doi.org/10.1007/s40846-020-00538-3.

·         W. Fadel, C. Kollod, M. Wahdow, Y. Ibrahim and I. Ulbert, "Multi-Class Classification of Motor Imagery EEG Signals Using Image-Based Deep Recurrent Convolutional Neural Network," 2020 8th International Winter Conference on Brain-Computer Interface (BCI), Gangwon, Korea (South), 2020, pp. 1-4, doi: 10.1109/BCI48061.2020.9061622.

·         C Zhang, Y.K. Kim, A Eskandarian. EEG-inception: an accurate and robust end-to-end neural network for EEG-based motor imagery classification. Journal of Neural Engineering, 2021, 18, 046014.

·         W. Huang, L. Wang, Z. Yan and Y. Liu, "Classify Motor Imagery by a Novel CNN with Data Augmentation," 2020 42nd Annual International Conference of the IEEE Engineering in Medicine & Biology Society (EMBC), Montreal, QC, Canada, 2020, pp. 192-195, doi: 10.1109/EMBC44109.2020.9176361.

·         Z Yu, W Chen, T Zhang, Motor imagery EEG classification algorithm based on improved lightweight feature fusion network, Biomedical Signal Processing and Control, 2022, 75, 103618, https://doi.org/10.1016/j.bspc.2022.103618.

·         Pérez-Zapata, A.F., Cardona-Escobar, A.F., Jaramillo-Garzón, J.A., Díaz, G.M. (2018). Deep Convolutional Neural Networks and Power Spectral Density Features for Motor Imagery Classification of EEG Signals. In: Schmorrow, D., Fidopiastis, C. (eds) Augmented Cognition: Intelligent Technologies. AC 2018. Lecture Notes in Computer Science, vol 10915. Springer, Cham. https://doi.org/10.1007/978-3-319-91470-1_14.

·         Y Han, B Wang, J Luo, L Li, X Li, A classification method for EEG motor imagery signals based on parallel convolutional neural network, Biomedical Signal Processing and Control, 2022, Volume 71, 103190, https://doi.org/10.1016/j.bspc.2021.103190.

4) The scope of the study is briefly explained at the end of the “Introduction” section in lines 86-90. However, the authors need to clearly express the novelty of the work and its contribution to the literature by discussing the research gap in the literature in detail. The references listed in 3) should be taken into account in explaining the novelty of the paper.

5) “Harr wavelet” on line 99 should be corrected as “Haar wavelet”.

6) The methodology section should be improved. I suggest the authors to revise the “Methods” section and present the algorithm in a detailed manner such as in the papers mentioned in 3).

7) A single case, namely 2008 BCI Competition IV Data Set 2a, is analyzed in the “Results” section. The quality of the paper can be improved by increasing the number of cases such as in:

·         Z Dokur, T Olmez, Classification of motor imagery electroencephalogram signals by using a divergence based convolutional neural network, Applied Soft Computing, 2021, 113, 107881, https://doi.org/10.1016/j.asoc.2021.107881.

·         Y Han, B Wang, J Luo, L Li, X Li, A classification method for EEG motor imagery signals based on parallel convolutional neural network, Biomedical Signal Processing and Control, 2022, Volume 71, 103190, https://doi.org/10.1016/j.bspc.2021.103190.

·         Shajil, N., Mohan, S., Srinivasan, P. et al. Multiclass Classification of Spatially Filtered Motor Imagery EEG Signals Using Convolutional Neural Network for BCI Based Applications. J. Med. Biol. Eng. 40, 663–672 (2020). https://doi.org/10.1007/s40846-020-00538-3.

·         W Ma, H Xue, X Sun, S Mao, L Wang, Y Liu, Y Wang, X Lin, A novel multi-branch hybrid neural network for motor imagery EEG signal classification, Biomedical Signal Processing and Control, 2022, 77, 103718, https://doi.org/10.1016/j.bspc.2022.103718.

·         Zhang, K.; Xu, G.; Han, Z.; Ma, K.; Zheng, X.; Chen, L.; Duan, N.; Zhang, S. Data Augmentation for Motor Imagery Signal Classification Based on a Hybrid Neural Network. Sensors 2020, 20, 4485. https://doi.org/10.3390/s20164485

8) Table 3 compares the accuracy of the proposed classification method with the methods presented in the literature. This table and the related discussion should be extended to include the results of papers mentioned in 3).

Reviewer 4 Report

This manuscript proposed a novel deep learning-based method for automated classification of motor imagery EEG signals, where convolutional neural network (CNN) was developed for the task of interest. In the proposed method, the time-frequency domain features were extracted from EEG signals, which were used as the inputs of CNN model. The performance of the proposed method has been validated using experimental data, with satisfactory result of 97.61% classification accuracy. Overall, the topic of this research is interesting, and the manuscript was well organised and written. The detailed comments are suggested as follows.

1.       Please clearly clarify the main contribution and innovations of this research in both abstract and introduction.

2.       Broaden and update literature review on CNN and its practical applications. E.g. Vision-based concrete crack detection using a hybrid framework considering noise effect.

3.       Please explain the advantage of CWT in signal analysis and feature extraction. Why not s-transform or HHT for getting time-frequency spectrum?

4.       The performance of CNN model is mainly related to network hyperparameters. How did the authors optimise hyperparameters to get the best classification accuracy?

5.       In the result comparison, please use some diagrams, such as confusion matrix.

6.       How about the robustness of the proposed method against noise effect?

7.       More future research should be included in conclusion part.

Round 2

Reviewer 3 Report

The manuscript can be published in the journal Sensors. The authors should make the following minor corrections:

1) Sentences that start with "And" on lines 17 and 90 should be corrected.

2) “morlet” on line 97 should be corrected as “Morlet”.

3) “…to transform the data. Which…” on line 108 should be corrected as “…to transform the data, which…”

4) The explanation presented for Eq. (4) on lines 178-180 should be corrected. Multiple definitions are given for k.

Author Response

Thank you for taking the time to review our manuscript and for providing valuable feedback. 

Here are responses to the reviewer comments:

Comment 1: Sentences that start with "And" on lines 17 and 90 should be corrected.

Reply1: corrected.

Comment 2: “morlet” on line 97 should be corrected as “Morlet”.

Reply 2: Done.

Comment 3: “…to transform the data. Which…” on line 108 should be corrected as “…to transform the data, which…”

Reply 3: Done.

Comment 4:  The explanation presented for Eq. (4) on lines 178-180 should be corrected. Multiple definitions are given for k.

Reply 4: We corrected the mistake and redefined k and other parameters for Eq. (4). 

Thank you again for your time and effort in reviewing our work.